# Entropy, Disagreement, and the Limits of Foundation Models in Genomics

**Maxime Rochkoulets**[1,2]     **Lovro Vrček**[1]     **Mile Šikić**[1,3]

[1]Genome Institute of Singapore, A*STAR, Singapore     [2]KU Leuven, Belgium
[3]Faculty of Electrical Engineering and Computing, University of Zagreb, Croatia

## Abstract

Foundation models in genomics have shown mixed success compared to their counterparts in natural language processing. Yet, the reasons for their limited effectiveness remain poorly understood. In this work, we investigate the role of entropy as a fundamental factor limiting the capacities of such models to learn from their training data and develop foundational capabilities. We train ensembles of models on text and DNA sequences and analyze their predictions, static embeddings, and empirical Fisher information flow. We show that the high entropy of genomic sequences—from the point of view of unseen token prediction—leads to near-uniform output distributions, disagreement across models, and unstable static embeddings, even for models that are matched in architecture, training and data. We then demonstrate that models trained on DNA concentrate Fisher information in embedding layers, seemingly failing to exploit inter-token relationships. Our results suggest that self-supervised training from sequences alone may not be applicable to genomic data, calling into question the assumptions underlying current methodologies for training genomic foundation models.

## 1 Introduction

Genomic Foundation Models (GFMs) represent a significant part of current research in machine learning applied to biology. This approach consists in training large deep learning models on vast collections of DNA sequences, using techniques such as autoregressive or masked language modeling, essentially training the model to predict unseen tokens from their surrounding context. The goal is that the model will eventually be able to learn hidden relations or grammar encoded in genomic sequences, and develop general biological underlying knowledge. This knowledge, materialized in its embeddings, can then be applied to a variety of downstream tasks. In natural language processing, this approach has been very successful, with transformer models such as BERT (Devlin et al., 2019) or RoBERTa (Liu et al., 2019) becoming widely used as a base for many NLP applications.

Despite numerous papers reporting good results on a variety of downstream tasks (Ji et al., 2021; Nguyen et al., 2024; Dalla-Torre et al., 2025), the performance and usefulness of GFMs trained in self-supervised settings from sequences alone has recently been questioned (Vishniakov et al., 2026). In fact, it has been observed that specialized solutions, such as smaller models trained specifically for the task, often outperform GFMs embeddings (Feng et al., 2025) while being faster and cheaper to train and much more convenient to use.

Having said that, no explanation for why these models perform poorly has been proposed so far. In this work, we argue that entropy provides a natural starting point, as human text and DNA sequences differ fundamentally from an information-theoretic perspective (Shannon, 1951; Schmitt & Herzel, 1997). We show that the low information content of DNA—from the point of view of unseen token prediction—prevents models from learning a confident and interpretable distribution, leads to disagreement between models, and causes models to not leverage inter-token relationships.

## 2 Methodology

We train 3 ensembles each made of $N$ transformer encoder BERT models (Devlin et al., 2019). The first ensemble is trained on English text with a byte-pair encoding tokenizer, the second on DNA

sequences, also using BPE tokenization, ensuring meaningful comparison with the text models, and a third ensemble is also trained on DNA, but uses a $k$-mer non-overlapping tokenizer, a more widely used tokenization scheme for genomic language models in practice (Dalla-Torre et al., 2025).

All $3N$ models use the exact same architecture and thus have an equal number of parameters (90M). They are trained on the same number of tokens (5B) and their vocabulary size is fixed to $4^6 = 4096$, matching a 6-mers tokenization scheme. Models iterate over the data in the same deterministic order, but their weights are randomly initialized. We used $N = 5$ in this work, that we consider a good tradeoff between number of models and training costs. Details regarding training procedure, datasets and hyperparameters can be found in Appendix A.1 and A.2.

Matching hyperparameters across (data, tokenizer) pairs allows us to compare the models precisely, and to show that fundamental differences between natural language and genomic sequences yield very different models, even when matched in data, architecture and training. We argue that these differences are best explained by high training data entropy.

## 3 ENSEMBLE DISAGREEMENT

Naturally, we expect the high entropy of DNA to be reflected in the output distributions of our models. We set the stage by computing the Kullback–Leibler (KL) divergence with the uniform distribution $\mathbb{E}_{x \sim \mathcal{D}} [D_{\mathrm{KL}} (P \parallel \mathcal{U})]$, where $P$ denotes a model's discrete distribution over its vocabulary for predicting the masked token with context x sampled from dataset $\mathcal{D}$.

We find that for DNA models using a BPE tokenizer, this quantity is roughly 3 bits, and less than 1 bit for $k$-mer models, while it is more than 10 bits for text models. This large KL difference already suggests that DNA models yield predictions that reflect greater uncertainty, whereas text models are highly confident in their predictions, diverging significantly from $\mathcal{U}$.

In the following sections, we show that these introductory results translate into disagreement and instability across DNA models, especially when compared to their counterpart text models.

### 3.1 OUTPUT DISTRIBUTIONS

To compare the output distributions of two models, we use the Jensen-Shannon distance:

$$d_{\mathrm{JS}}(P_i, P_j) = \sqrt{\mathrm{JSD}(P_i \parallel P_j)} = \sqrt{\frac{1}{2} D_{\mathrm{KL}}(P_i \parallel M) + \frac{1}{2} D_{\mathrm{KL}}(P_j \parallel M)}$$

where $\mathrm{JSD}(P_i \parallel P_j)$ is the Jensen-Shannon divergence between two models $i \neq j$ of the same family and $M$ is the mixture distribution $M = (P_i + P_j) / 2$. The Jensen-Shannon distance defines a proper metric and is therefore symmetric. This makes it more suited than the KL divergence for measuring pairwise models agreement in terms of their output distributions.

At first glance, we find that DNA models seem to agree in their output distributions over unseen tokens as much as text models. However, the expected pairwise distance between DNA models increases when applying nucleus sampling (Holtzman et al., 2020), that is, reweighting the distribution by keeping only the top-$p$% of the probability mass.

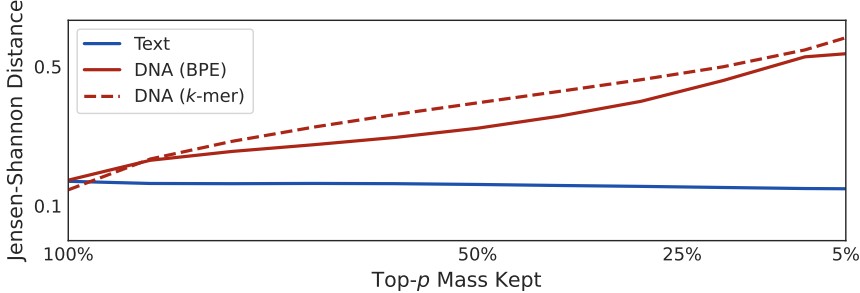

Figure 1: Jensen-Shannon distance $\mathbb{E}_{x \sim \mathcal{D}} [d_{\mathrm{JS}}(P_i, P_j)]$ between models of the same ensemble as a function of top-$p$ mass kept. Values computed over $100\,000$ samples unseen during training.

This shows that small distances for high values of $p$ are a consequence of the DNA models assigning roughly the same probability to each token, rather than agreeing over what the unseen token really is. Figure 1 shows that this apparent agreement breaks as $p$ is lowered. Expected distance between text models, on the other hand, is barely impacted as a function of $p$, indicating stable convergence in their output distributions, and strong inter-model agreement.

## 3.2 STATIC WORD EMBEDDINGS

Having shown that DNA models, unlike text models, disagree in their distributions over masked tokens, we aim to go deeper and evaluate their levels of agreement in embedding space. In this work, we limit ourselves to the first embedding layer, also called word embedding layer or static embeddings, in BERT terminology[1].

Although contextual representations are typically preferred when evaluating transformer encoder models, we show that our text models encode meaningful static relationships. This approach to word embeddings can be associated to older static embedding approaches such as word2vec (Mikolov et al., 2013). We included relevant examples showing that text models learned meaningful static associations in Appendix A.3.

To quantify ensemble disagreement at static word embeddings level, we first compare the relative positions of tokens within each pair of model, using top-$k$ Jaccard overlap and Spearman correlation of nearest-neighbors (both with $k = 10$). Spearman correlation is computed from both models and averaged. Results of local metrics with more values for $k$ can be found in Appendix A.4.

Then, we compare embeddings between each pair of models of the same ensemble. Local metrics are straightforward to compute, but comparing two models trained independently first requires aligning their embeddings via Procrustes (Schönemann, 1966). We report disparity, that is the sum of squares of the point-wise differences between the two aligned spaces, as well as cosine similarity, as it is standard in static word embeddings literature (Hamilton et al., 2016; Lample et al., 2018).

| | Top-10 Overlap ↑ | Local Spearman ↑ | Procrustes Cosine ↑ | Procrustes Disparity ↓ |
|---|---|---|---|---|
| Text | **0.54** | **0.68** | **0.71** | **0.49** |
| DNA (BPE) | 0.39 | 0.48 | 0.61 | 0.62 |
| DNA ($k$-mer) | 0.41 | 0.47 | 0.59 | 0.66 |

Table 1: Different measures of agreement in static word embedding space.

We emphasize the fact that models were trained on the same data, but also that all models of the same ensemble iterated over the dataset in the same deterministic order. Consistent differences between text and DNA models in Table 1 show that high data entropy, coupled with random weights initialization, leads to significantly more disagreement in DNA models than in text models.

Intuitively, this is due to the fact that in high entropy data, conditional distributions are flatter, with more tokens plausible at each position, making two models trained independently less likely to converge to the same static word embeddings state. Observing high variance across random seeds but same data indicates poor robustness to initialization and stochastic optimization. This has practical implications, reducing reproducibility and making static embedding based analyses unreliable.

## 4 EMPIRICAL FISHER INFORMATION

BERT-like models are typically made of static embedding layers, that map one-hot encoded tokens to dense representations, followed by transformer layers, that consider asymmetric inter-token relationships using attention (Vaswani et al., 2017), with an optional task-depending prediction head (Devlin et al., 2019; Rogers et al., 2020). In this section, we show that computing Fisher information in the light of each layer's function can provide an important clue to explain the poor performance of DNA foundation models embeddings on downstream tasks.

---

[1]To avoid any confusion with static embeddings created from contextual ones, also common in the literature, we use the term *static word embeddings* to characterize this layer.

Fisher information is a way to quantify the information content of the data with respect to a parameter of our model. When dealing with multiple parameters, the empirical Fisher information matrix (FIM) of parameters vector $\theta$ under dataset $\mathcal{D}$ is defined as follows[2]:

$$\tilde{F}(\theta) = \mathbb{E}_{(\mathbf{x},\mathbf{y})\sim\mathcal{D}} \left[ \nabla_\theta \log P(\mathbf{y} \,|\, \mathbf{x}, \theta) \, \nabla_\theta \log P(\mathbf{y} \,|\, \mathbf{x}, \theta)^T \right]$$

Here, we are only interested in the diagonal of the FIM, which doesn't require tracking the entire matrix. The FIM, through $\tilde{F}_{ii}(\theta)$, gives us an idea of how strongly parameter $\theta_i$ is constrained by the data, by measuring how sensitive is the log-likelihood to changes in that parameter.

By grouping each BERT layer into three groups, namely embeddings, transformer layers, and head, we can get an idea of the information content of a sample with respect to each group of layers. To do that, we sum the Fisher information of layers of the same group (averaged over all models of the same ensemble) and normalize the results. This layer-wise aggregation of empirical Fisher information reveals completely different distributions in text and DNA models.

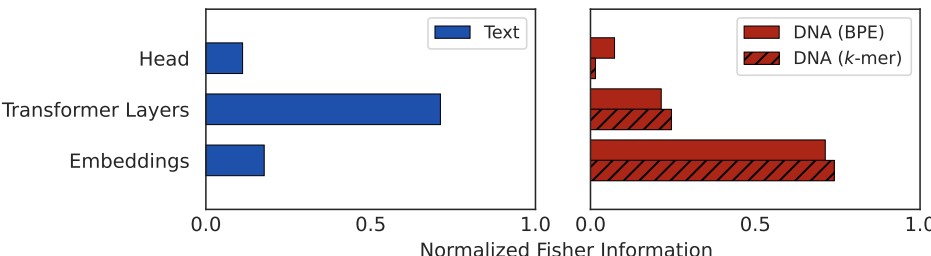

Figure 2: Normalized layer-wise aggregation of empirical Fisher information in text and DNA models. Estimate computed over $100\,000$ samples unseen during training.

We find that for text models, as one could have expected, Fisher information is concentrated in transformer layers, responsible for computing meaningful relationships between tokens (Figure 2). Strangely, a very different concentration of Fisher information appears in DNA models, where static embedding layers dominate. This could be an indication that DNA models do not look at inter-token relationships, and do not leverage the attention mechanism, even though it is a core component of their architecture. These results also point at memorization, where DNA models seem to be constrained by token identity, while text models care about their relationships. A complete plot showing all models and layers in included in Appendix A.5.

## 5 CONCLUSION

Following recent work that highlighted the limited performance of genomic foundation models, we aimed to go beyond empirical results and investigate possible explanations. Using entropy as a starting point, we performed a simple experiment consisting in comparing BERT models trained on English text to identical models trained on DNA sequences.

We proposed two main directions that we believe deserve deeper exploration in future work. First, high data entropy leads to uncertain predictions, but also to disagreement among models, even under matched training methodology, data and architecture. Second, analysis of aggregated empirical Fisher information suggests that DNA models seemingly fail to effectively capture inter-token relationships, as information is concentrated in static embeddings rather than in transformer layers. We also find that using different tokenization schemes for DNA had little impact in all of the reported metrics, further suggesting that our results reflect fundamental properties of the data itself.

Finally, our observations and results, coupled with previous critical work on genomic foundation models, suggest that self-supervised training on unseen token prediction from genomic sequences alone may be insufficient for models to develop foundational capabilities. Our results are fully reproducible using the source code available at `https://github.com/lbcb-sci/GFMs`.

---

[2]Although this definition of the empirical Fisher has been criticized in the context of natural gradient optimization (Kunstner et al., 2019), our use of it here is about relative distribution of Fisher mass across layers, not as a reliable tool for optimization.

MEANINGFULNESS STATEMENT

This work helps us learn meaningful representations of life because it shows that human text and DNA are fundamentally different from the point of view of unseen token prediction, the current dominant self-supervised training paradigm. More broadly, we hope our work calls into question if the assumptions we borrowed from NLP really transfer to genomics, or if the high entropy of the code of life breaks the tools and models we built for processing the language of humans.

ACKNOWLEDGEMENTS

This work was supported by the A*STAR Genome Institute of Singapore (A*STAR GIS).

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

## A  APPENDIX

### A.1  MODELS CONFIGURATION

We used the default BertConfig object from HuggingFace's Transformers library and only changed the vocabulary size to $4096$, leaving the rest of the hyperparameters to their default values. All BERT models use the same configuration which results in models made of 90M trainable parameters.

The detailed list of default hyperparameter values can be viewed at:
https://huggingface.co/docs/transformers/en/model_doc/bert

### A.2  TRAINING PROCEDURE AND DATASETS

All models are trained on a total of 5.12B tokens using the AdamW optimizer. The learning rate is set to $1 \times 10^{-4}$ and max gradient norm to $0.5$. The training lasts 5 epochs, each epoch iterates over 2M sequences of 512 tokens, padded or truncated if necessary. We use standard masked language modeling settings, masking 15% of tokens randomly in each sample. Training was done on Nvidia A100-40GB GPUs with a batch size of 96. The data is fed to all models in the same order (same data seed), but each model is initialized with a different random seed.

For training text models, we used the Wikipedia dataset published by the Wikimedia foundation:
https://huggingface.co/datasets/wikimedia/wikipedia

For training DNA models, we used plant reference genomes sequences from the Zhang Tao Lab:
https://huggingface.co/datasets/zhangtaolab/plant-reference-genomes

### A.3  STATIC WORD EMBEDDINGS EXAMPLES

The following are examples that our text models effectively grouped tokens in meaningful ways in their static word embedding space. We pick a token representing a full word and print its 5 closest neighbors in static word embeddings using cosine similarity:

- Closest tokens from [America]: [Africa, Europe, Britain, Canada, Carolina]
- Closest tokens from [football]: [baseball, basketball, Football, hockey, occer]
- Closest tokens from [France]: [Spain, Italy, Paris, Germany, French]
- Closest tokens from [computer]: [technology, video, tware, data, systems]
- Closest tokens from [students]: [schools, parents, children, studies, programs]

These examples are extracted from a single text model, but we verified that all text models had sound relationships in their embedding space, which is also supported by the results shown in Table 1. This should validate that looking at static word embeddings to compare models makes sense, although further work should also look into contextual embeddings and layers activations.

## A.4 ADDITIONAL STATIC WORD EMBEDDINGS RESULTS

|  | Top-1 | Top-3 | Top-5 | Top-10 | Top-20 | Top-50 | Top-100 | Top-1000 | Top-4000 |
|---|---|---|---|---|---|---|---|---|---|
| Text | **0.75** | **0.60** | **0.56** | **0.54** | **0.52** | **0.50** | **0.49** | 0.52 | **0.97** |
| DNA (BPE) | 0.42 | 0.38 | 0.38 | 0.39 | 0.40 | 0.41 | 0.41 | 0.52 | **0.97** |
| DNA (k-mer) | 0.37 | 0.36 | 0.38 | 0.41 | 0.42 | 0.41 | 0.40 | **0.54** | 0.96 |

Table 2: Additional results for top-$k$ overlap.

|  | $k=3$ | $k=5$ | $k=10$ | $k=20$ | $k=50$ | $k=100$ | $k=1000$ | $k=4000$ |
|---|---|---|---|---|---|---|---|---|
| Text | **0.68** | **0.68** | **0.68** | **0.67** | **0.65** | **0.65** | **0.61** | 0.73 |
| DNA (BPE) | 0.39 | 0.44 | 0.48 | 0.51 | 0.54 | 0.55 | 0.57 | **0.76** |
| DNA (k-mer) | 0.34 | 0.40 | 0.47 | 0.53 | 0.57 | 0.56 | 0.58 | **0.76** |

Table 3: Additional results for Spearman nearest-neighbor correlation.

## A.5 FULL FISHER INFORMATION

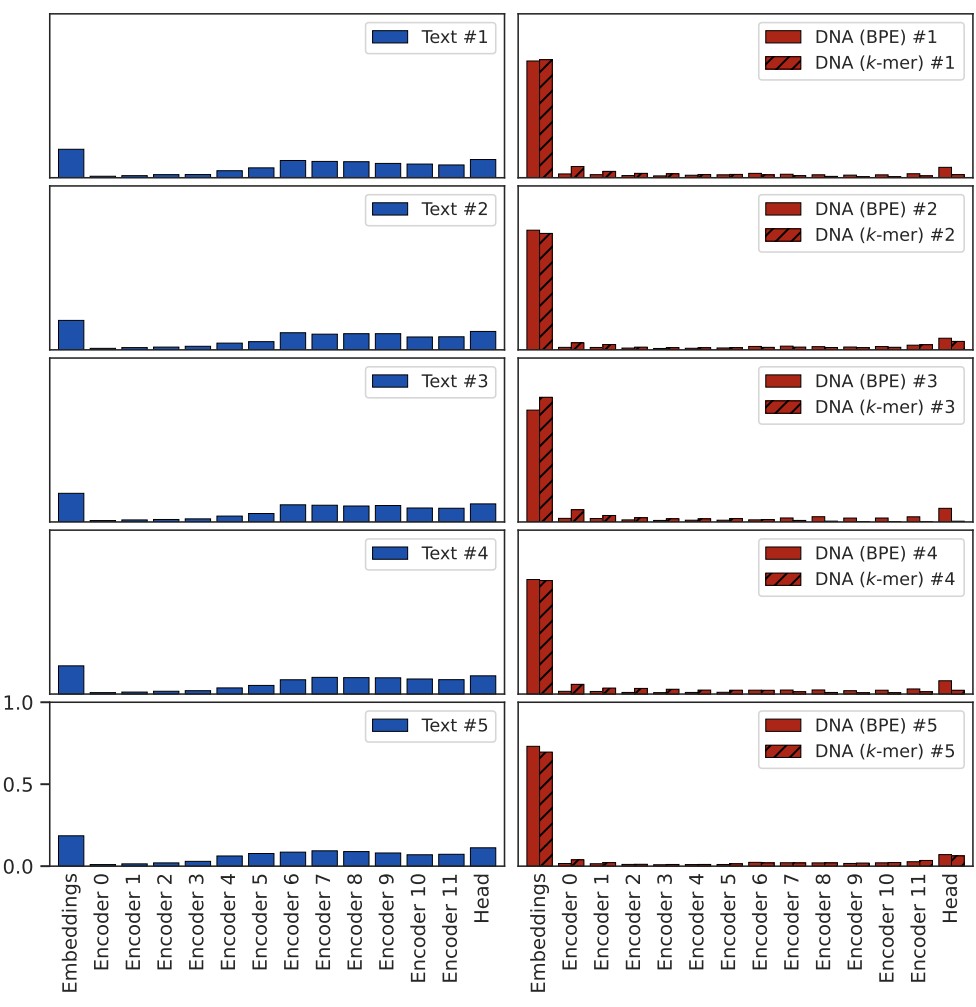

Figure 3: Detailed normalized empirical Fisher information content per model, for each layer. Estimate computed over $100\,000$ samples unseen during training.

