# OpenReview forum: "Entropy, Disagreement, and the Limits of Foundation Models in Genomics"
_ICLR.cc/2026/Workshop/LMRL — ICLR 2026 Workshop LMRL Poster_

### Official Review · Reviewer_Bbr7 · 2026-02-11
**DNA region-specific analysis not considered**

**Rating:** 6
**Confidence:** 4

**Review:**

The author's compare the agreement between ensembled, BERT-based LLMs trained on either natural language or genomic sequence modalities. Evaluating the strengths and weaknesses of genomic foundation models in an information-theoretic perspective is reasonable and a potentially valuable research direction.

However, a major oversight of this work is that genomic sequences are not considered with respect to specific, well-defined genomic regions (promoters, enhancers, UTRs coding regions, centromeres, telomeric regions etc.), nor are any of these regions mentioned in the manuscript. I highly suspect the author's results would differ if applied to well conserved regions of the genome that have frequently occurring motifs (like codons).

In practice, DNA encodes information that is reliably read and interpreted by the cellular machinery. It is therefore not feasible that the genome is inherently entropic. Instead, it seems more likely that model architecture, capacity and data considerations are more likely to explain the suggested limitations of genomic foundation models.

---

### Official Review · Reviewer_GCDK · 2026-02-19
**An information-theoretic lens on GFMs limitations: compelling diagnostics, but scale and generality remain open questions**

**Rating:** 6
**Confidence:** 3

**Review:**

The reviewer would like to thank the authors for their work. Please find below strengths, weaknesses and clarifications / questions.

## Summary

This paper tackles the question of GFMs' poor performance through the lens of information theory and statistical analyses. By comparing ensembles of text and DNA models under controlled experimental conditions, it shows via several analyses (alignment of output distributions in MLM, alignment of static word embedding spaces, localisation of Fisher information) that intrinsic differences between DNA and textual data (in particular the high entropy of genomic sequences) may explain this performance gap.

## Strengths

**Clean experimental design.** The comparison is well controlled: same architecture (90M params), same token count (5B), same vocab size (4096), same data ordering, with only the data and random initialization differing. This isolates the effect of the data itself. The inclusion of both BPE and k-mer tokenization, with consistent results across both, preempts the objection that findings could be a tokenization artifact.

**Novel explanatory angle.** Most prior work shows empirically that GFMs underperform. This paper instead asks why and compare DNA and text models through three complementary lenses: (i) the agreement of output distributions under MLM across independently trained models, (ii) the agreement of static word embeddings across independently trained models, and (iii) where Fisher information concentrates across layers, revealing which parts of the architecture are actually constrained by the data. This shift from benchmarking to mechanistic understanding is welcome.

**Fisher information analysis.** The observation that DNA models concentrate Fisher information in embedding layers while text models concentrate it in transformer layers is interesting, and as far as I know, new. It suggests that DNA models are not meaningfully using the attention mechanism, which could have implications for future architecture and training choices.

**Reproducibility.** Source code is provided, experimental details are thorough (Appendix A.1–A.2), and the setup is modest enough (90M params, A100 GPUs) that others could realistically replicate.

## Weaknesses

**Small scale.** 90M parameters and 5B tokens is far below the scale of real GFMs (e.g., Nucleotide Transformer at 2.5B). A natural counter-argument is that at sufficient scale, the weak signal in DNA could become learnable. The paper does not address this.

**Plant genomes only.** The DNA models are trained on plant reference genomes, yet the GFM literature is primarily about human genomics. Plant genomes have different repeat structure, GC content, and organization. This choice is not justified and it is unclear whether the conclusions transfer to human or multi-species settings.

**Data quality asymmetry.** Wikipedia is curated, well-edited, information-dense text. Plant reference genomes are raw sequence. These are not equivalent data sources. The paper's argument is that this asymmetry is the point (DNA is inherently higher entropy), but it should at least be acknowledged and discussed. A control using low-quality or repetitive text would help disentangle data quality from intrinsic entropy.

**No confidence intervals or error bars.** None of the results in Tables 1–3 or Figures 1–2 include confidence intervals, standard deviations, or error bars. It is hard to judge whether the observed differences between text and DNA models are statistically robust.

## Clarifications / Questions

**Spearman computation is not clearly explained.** The procedure for computing Spearman nearest-neighbor correlation is underspecified. Is the top-k neighbor set taken from one model only (making the metric asymmetric), or from the intersection of both models' sets? If asymmetric, is the result averaged over both directions? How are tokens that appear in one model's top-k but not the other's handled? These details matter for interpretation and reproducibility.

**Missing related work.** Some relevant papers are not cited. Zhang et al. (2023, https://academic.oup.com/bioinformatics/article/39/10/btad617/7303863) showed that k-mer embeddings pretrained on random DNA match the performance of those trained on real genomes, which directly supports the entropy argument. Marin et al. (2023, BEND benchmark, https://arxiv.org/abs/2311.12570) found that GFMs capture limited long-range information, which connects to the Fisher information finding. These references would strengthen the paper's positioning.

---

### Official Review · Reviewer_fpyP · 2026-02-24
**-**

**Rating:** 6
**Confidence:** 4

**Review:**

Summary:

The authors investigate the limited performance of Genomic Foundation Models (GFMs) compared to their NLP counterparts, by comparing identical BERT models trained on textual sequences vs. DNA sequences. By isolating entropy and weight initialization, they find that DNA models exhibit near-uniform output distributions, high cross-seed disagreement in static embeddings, and a higher concentration of empirical Fisher information in the initial embedding layers rather than the transformer blocks.


Strengths:
- I believe that the application of empirical Fisher information to diagnose layer-wise information bottlenecks in GFMs is highly novel. It’s aim to evaluate biological representations is very relevant, and directly addresses the workshop’s call for evaluating methods and limits for this biological modality.
- For a tiny paper, the experimental setup is impressively controlled – by matching the architecture / data / vocab size / training order, the authors seem to successfully isolate the impact of entropy on the models’ convergence.
- The core hypothesis and empirical findings are concise and clear, and each claim is followed up by a valid empirical “proof”.

Weaknesses:
- I question whether the strict setting of the vocab size as 4K is the right call. While the reason for doing so is understandable (matching hyperparameters for control), it “crushes” the text vocabulary and may force the BPE tokenizer into sub-word or character-level splits that quite likely inflate the text model's predictability due to deterministic spelling completion rather than true semantic prediction – making the comparison to the high-entropy DNA somewhat skewed.
- The authors claim that "no explanation for why these models perform poorly has been proposed so far" and generalize findings from a vanilla BERT model to question general self-supervised learning on genomic sequences. This feels to me like ignoring existing literature and newer, non-BERT architectures, or even other research tackling the same exact questions.
- The motivation relies on the poor downstream performance of GFMs, but the authors do not empirically test if their specific findings actually correlate with downstream task degradation or show any downstream task effect at all.

I would advise the authors to soften their strong claims, explicitly acknowledging that the insights from the paper apply primarily to the vanilla BERT architecture trained via MLM. Also, briefly discussing how much of the text model's high KL divergence is driven by deterministic sub-word spelling would add necessary nuance to the cross-modality comparison.

Overall, the highly controlled experimental methodology and novel application of empirical Fisher information make this, in my eyes, an insightful work-in-progress for the workshop, provided the authors dial back their sweeping claims.

---

### Meta-Review · Area_Chair_A8mY · 2026-02-25

**Recommendation:** Accept (Poster)
**Confidence:** 4

**Metareview:**

Accept

---

### Decision · Program_Chairs · 2026-03-02

**Decision:**

Accept (Poster)

**Comment:**

Please see the meta-review.